# Transient Increases in Inflammation and Proapoptotic Potential Are Associated with the HESN Phenotype Observed in a Subgroup of Kenyan Female Sex Workers

**DOI:** 10.3390/v14030471

**Published:** 2022-02-25

**Authors:** Marcel Gluchowski, Xiaoqiong Yu, Bernard Abrenica, Samantha Yao, Joshua Kimani, Renée N. Douville, Terry Blake Ball, Ruey-Chyi Su

**Affiliations:** 1Department of Biology, University of Winnipeg, Winnipeg, MB R3B 2R9, Canada; marcel.gluchowski@gmail.com (M.G.); r.douville@uwinnipeg.ca (R.N.D.); 2Yangquan Municipal Center for Disease Prevention and Control, Yangquan 045000, China; yxq7017@163.com; 3National HIV & Retrovirology Laboratories, Public Health Agency of Canada, Winnipeg, MB R3E 3L5, Canada; bernard.abrenica@phac-aspc.gc.ca (B.A.); t.blake.ball@phac-aspc.gc.ca (T.B.B.); 4Department of Immunology, University of Manitoba, Winnipeg, MB R3E 0T5, Canada; samanthayao1031@gmail.com; 5Medical Microbiology & Infectious Diseases, University of Manitoba, Winnipeg, MB R3E 0J9, Canada; jkimani@csrtkenya.org; 6Department of Medical Microbiology, University of Nairobi, Nairobi 00202, Kenya

**Keywords:** HIV-1, HESN, ISG, PBMC, IFN-γ, antiviral, commercial sex workers

## Abstract

Interferon (IFN) -stimulated genes (ISGs) are critical effectors of IFN response to viral infection, but whether ISG expression is a correlate of protection against HIV infection remains elusive. A well-characterized subcohort of Kenyan female sex workers, who, despite being repeatedly exposed to HIV-1 remain seronegative (HESN), exhibit reduced baseline systemic and mucosal immune activation. This study tested the hypothesis that regulation of ISGs in the cells of HESN potentiates a robust antiviral response against HIV. Transcriptional profile of a panel of ISGs with antiviral function in PBMC and isolated CD4+ T cells from HESN and non-HESN sex worker controls were defined following exogenous IFN-stimulation using relative RT-qPCR. This study identified a unique profile of proinflammatory and proapoptotic ISGs with robust but transient responses to exogenous IFN-γ and IFN-α2 in HESN cells. In contrast, the non-HESN cells had a strong and prolonged proinflammatory ISG profile at baseline and following IFN challenge. Potential mechanisms may include augmented bystander apoptosis due to increased *TRAIL* expression (16-fold), in non-HESN cells. The study also identified two negative regulators of ISG induction associated with the HESN phenotype. Robust upregulation of *SOCS-1* and *IRF-1*, in addition to *HDM2*, could contribute to the strict regulation of proinflammatory and proapoptotic ISGs in HESN cells. As reducing IRF-1 in the non-HESN cells resulted in the identified HESN ISG profile, and decreased HIV susceptibility, the unique HESN ISG profile could be a correlate of protection against HIV infection.

## 1. Introduction

Despite the tremendous efforts put forth, the number of new HIV infections remained stable at around 1.3 million per year for the last 3 years [1,2,3]. The success of preventative measures, such as increased condom use, pre-exposure prophylaxis (PrEP), and treatment as prevention (TasP) remain suboptimal and dependent on the level of adherence [4,5,6,7,8,9]. There is an ever-urgent need for novel preventative tools, which require better understanding of the protective mechanisms against HIV-1 infection.

Fortunately, not all individuals exposed to HIV-1 become infected. Several studies reported a subset of around 5–15% of HIV-exposed individuals who remain seronegative (HIV-exposed seronegative (HESN)) even after repeated exposure [10]. These HESN individuals have been found in several geographic locations with diverse ethnic backgrounds and routes of exposure. Studies have been performed to identify genetic and immunologic mechanism(s) of protection in the HESN subsets [11,12,13,14,15,16,17,18,19,20,21,22,23,24,25,26,27,28,29,30,31,32,33,34,35,36]. The mechanism(s) reported to be associated with seeming resistance to HIV infection in HESN are most likely multifactorial [16,26,36,37]. Host factors that are associated with delayed seroconversion or reduced susceptibility to HIV infection include *CCR5, CCR2b, ERAP2, MX2, TIM3, TLR3, APOBEC3* family, *TRIM* family, *HLA (DQB1*03:02, HLA-B*57 and HLA-B*35, HLA-E, HLA-F, CD1a-e, MR1*), Vitamin D receptor (*VDR*), and *IRF-1*. In addition to these genetic factors, a generalized phenotype of immune quiescence (IQ) has also been observed in HESN [26]. IQ is characterized by reduced baseline expression of T-cell activation markers, low baseline levels of pro-inflammatory cytokine and chemokine production in the peripheral and genital mucosal of the HESN, and increased baseline numbers of systematic regulatory T cells (Treg), which are known to suppress T-cell activation [38]. Several genetic factors also support the IQ phenotype in HESN; for example, reduced baseline interferon (IFN) regulatory factor-1 (IRF-1) expression and increased baseline VDR expression, which are associated with reduced immune activation in HESN [14,25,39,40,41]. Our earlier studies characterized IRF-1 in a highly HIV-exposed commercial sex worker cohort in Nairobi, Kenya. IRF-1, an IFN-stimulated gene (ISG) is a functional correlate of protection against HIV-1 acquisition [14,41,42,43] and its expression is regulated by both genetic and epigenetic mechanisms in HESN. This study further investigated the potential role of IRF-1 in supporting the IQ phenotype in HESN, perhaps contributing to reduced HIV susceptibility.

The IFN signaling pathway protects cells from invading viral pathogens by trans-activating a variety of interferon-stimulated genes (ISGs) [44,45,46,47]. Some ISGs encode innate detectors of viral molecules, while some encode effectors with varied antiviral functions, such as regulators of the apoptotic pathway, and IFN-signaling molecules [45,48,49]. Our studies and others suggest that a “fine-tuned” ISG response is critical for suppressing undesired, exaggerated viral responses, including HIV, that induce prolonged immune activation, and for preventing the consequential pathogenic host damages, while maintaining some innate antiviral activity [41,43,44,46]. Persistent type-I interferon responses have been associated with pathogenic outcomes of HIV-1 and SIV infections [46,50,51,52]. Our earlier work found that CD4+ T cells from Kenyan HESN female sex workers (FSWs) exhibit a lower basal level of IRF-1 but have the capacity for normal and transient *IRF-1* response to HIV-1 infection. In vitro studies that moderate reduction of cellular *IRF-1* expression via small-interfering RNA (siRNA) by 30–60%, decreased in vitro susceptibility to HIV-1 infection by >95% [14,41,43,53,54,55]. Although IRF-1 is an ISG itself, it is also a key transcription regulator for the ISG transcriptional program [56,57,58]. Thus, IRF-1 is a key regulator of late antiviral ISGs. Together, these support the crucial role of modulating ISG-mediated host defenses/inflammation in determining the outcome of infection.

This study focused on examining the expression and regulation of a subset of ISGs that have the described antiviral function. It tested the hypothesis that antiviral ISGs are uniquely regulated in immune cells from HESN individuals and that strict regulation of these ISGs is strongly associated with protection against acquisition of HIV-1 infection observed in HESN.

Furthermore, expression analysis of ISGs revealed that during both acute and chronic viral infection, most differentially expressed ISGs are induced by both type-I and type-II IFNs [59,60]. In addition, these data suggest that IFN-γ strongly contributes to shaping ISG upregulation, in addition to type-I IFNs. This study thus also compared the response of key ISGs to IFN-γ and IFN-α2 stimulation in the PBMC from HESN and non-HESN individuals. A unique set of proinflammatory antiviral ISGs and proapoptotic ISGs, which mounted normal but brief transcriptional responses, were found to have close association with the low susceptibility to HIV infection observed in HESN FSWs.

## 2. Materials and Methods

### 2.1. Ethics

The ethical review boards from the Kenyatta National Hospital/University of Nairobi (#P211/09/2006, renewed and amended annually since 2006), the University of Manitoba (HS15280-B2012:043, regularly renewed since 2012), and the Regional Ethical Review Board in Stockholm (2014/959–31, amended in 2018: 2018/1306–31) approved the study. Informed written consents were obtained from the participants in this study, following the declaration of Helsinki protocols.

### 2.2. Study Subjects

Participants with natural resistance to HIV infection (HESN) and HIV-susceptible seronegative participants (non-HESN) were chosen semi-randomly from subsets (age ranging 21–46 years at the time of sample collection, >7 or <3 years in the sex trade, active in sex work the week prior to sample collection and having no known infections, seronegative for HIV-1) of a well-characterized cohort of FSWs at the Majengo Clinic, Pumwani district of Nairobi, Kenya during the years of 2006–2010 (Table 1). These women maintained high-risk sexual behavior (0–70 clients the week prior to sample collection) with known HIV-infected clients (ranging between 0–3 clients). HESN and non-HESN FSWs were HIV negative (tested by ELISA and DNA/RNA testing) for >7 years and <3 years of follow-up, respectively. The non-HESN participants had been active in the sex trade for <3 years and were seronegative when the blood samples were collected. The definition of HESN and non-HESN is based on available epidemiological data of this cohort [26,61,62]. All FSWs participating in the Majengo cohorts were followed up twice a year during resurveys, which ended in 2013. By 2013, all HESN participants in this study remained seronegative for HIV-1, despite unprotected sexual behaviors. By 2013, 11 of the non-HESN participants (42%) seroconverted; 8 of the non-HESN participants (31%) did not return for follow-ups after 2009/2010. The other non-HESN FSWs (*n* = 7) remained seronegative up to the 2011/2012 resurvey. Despite an earlier study of this cohort demonstrating a ~85% seroconversion rate among the FSWs attending the Majengo clinic during a 13–14-year follow-up period [61], the HIV incidence has greatly dropped in Nairobi [63], perhaps due to the huge rollout of antiretroviral treatments (ART) in 2009 and active educational campaigns for safe sex and condom use. However, with the relatively small biological variations and significant differences observed within each group in the parameters examined in this study, the potential caveat pertaining to the epidemiological definition of HESN and non-HESN is deemed acceptable for this study.

### 2.3. Cell Culture

PBMCs were isolated by Ficoll–Hypaque density gradient centrifugation, frozen, and shipped to University of Manitoba where experiments were performed. Upon thawing (cell viability > 90%), PBMCs were cultured immediately in RPMI 1640 (supplemented with 10% fetal bovine serum, 100 unit/mL of penicillin, and 100 μg/mL of streptomycin) for 3 h at 37 °C and 5% CO_2_. Resting CD4^+^ T cells were enriched using StemSep^®^ Human Naïve CD4^+^ T Cell Enrichment Kit (STEMCELL Technologies, Vancouver, BC, Canada), following the manufacturer’s protocol. The enriched population was CD4^+^ when assessed on a flow-cytometer (>95% CD3^+^CD4^+^). The PBMCs and CD4+ T cells (after an additional 3 h of resting, to minimize in vitro cell activation as a result of the enrichment process) were then stimulated with IFN-γ (10 ng/mL) or IFN-α2 (10 ng/mL; Sigma, Oakville, ON, Canada) for the indicated time.

### 2.4. Synthesis and Purification of CD4-Binding Aptamer-siRNA Chimeras (CD4-AsiCs)

Chimeric RNAs, composed of an siRNA (specific of *IRF-1* or a scrambled negative control) fused to a CD4-aptamer, a structured RNA selected to bind to surface CD4 molecules with high affinity, were used to knock down the expression of *IRF-1* in CD4+ T cells, since CD4+ T cells were the main target cells for HIV-1 infection. CD4-AsiCs were generated following the protocol established in Dr. Lieberman’s laboratory [64]. Briefly, CD4-AsiCs were designed with a CD4 aptamer at the 5′ end ligated to the sense (inactive) siRNA strand. The construct was cloned into an expression plasmid and was in vitro transcribed, using the 2′-fluoropyrimidines to enhance stability in culture and reduce stimulation of cellular immune sensors that detect foreign nucleic acid. CD4-AsiCs transcripts were eluted from denaturing SDS-PAGE gels and analyzed by native SDS-PAGE gels, before annealing the antisense (active) siRNA strand. IRF-1 siRNA (3′ IRF-1 siRNA: 5′ p- aag gaU gcc UgU UUg UUc cgg dTdT -3′) and scrambled siRNA (3′ scrambled siRNA: 5′ p- aaU UcU ccg aac gUc Uca cgU dTdT -3′) were ordered from Integrated DNA Technologies (IDT). CD4+ T-cell-enriched cultures were incubated with 5μM of *IRF-1*-specific or scrambled-control CD4-AsiCs for 16 h at 37 °C, 5% CO_2_. The knockdown of IRF-1 expression was quantitated using intracellular staining and flow cytometry with antibodies specific for IRF-1 (ab243895, Abcam). Cell cultures with confirmed *IRF-1* knockdown (30–60%) were then treated with media or IFN-*γ* (10 ng/mL) as described in the text.

### 2.5. Quantitative RT–PCR

Cells were lysed in buffer LTR (Qiagen, Venlo, The Netherlands) with β-mercaptoethanol and total RNA was isolated from lysates as described previously [41], using Trizol (Sigma, Oakville, ON, Canada) and RNeasy MinElute Cleanup Kit (Qiagen, Toronto, ON, Canada). RNA was treated with RNase-free DNase I prior to reverse transcription (Qiagen, Toronto, ON, Canada). Resulting cDNA was evaluated in qPCR with specific primer sets for listed ISGs (Appendix A), and 18S rRNA (sequences available upon request). Annealing temperature for all primer sets was 60 °C. All qPCR were performed with SYBR-Green qPCR Master Mix (Qiagen, Toronto, ON, Canada). All primer sets used in the study were tested for amplification efficiencies and the results were similar (Appendix A). Average threshold cycle (Ct) from duplicate wells (with covariance less than 10%) was determined and standardized with the 18S rRNA internal control (input control) for relative expression level and normalized to untreated, culture media (CM) -alone culture conditions (as a reference) using a comparative ∆∆Ct program (LightCycler 480 Real-Time PCR System; Roche Applied Science, Laval, QC, Canada) for fold-changes in transcript level.

### 2.6. Data Analysis

Statistical analyses were performed with Graph Pad Prism 9.0 (San Diego, CA, USA). Normality tests were performed for each sample set. Data sets assuming Gaussian distribution were analyzed using parametric statistical tests; non-Gaussian distributed sample sets were analyzed using nonparametric statistical tests. The unpaired *t*-test was used to determine whether mean/median values differed significantly between two groups of sample sets. One-way analysis of variance was used when more than two groups of data sets were involved in the analysis.

## 3. Results

### 3.1. The PBMC of HESN FSWs Have Low Levels of ISG Transcripts Encoding Proinflammatory Signalling Pathway Proteins but Increased Transcripts of 2 ISGs, Key to the Detection of RNA and DNA Viruses

To determine whether HESN FSWs have constitutively stronger basal antiviral defenses, transcripts of the genes encoding ISGs with antiviral function were assessed using relatively quantitative RT-PCR. A total of 76 ISGs that have been shown to play crucial roles in antiviral immunity [45,47,56,58,65,66] were examined in the peripheral blood mononuclear cells (PBMC) from HESN women and non-HESN FSWs (Table 2, Appendix A). Only 35 of the 76 ISGs examined had detectable RNA transcripts in ex vivo, PBMC (*n* = 16).

The other 41 ISGs, although not detected via RT-qPCR in unstimulated PBMC, might still be differentially regulated in HESN PBMCs upon viral infection or challenge with exogenous stimuli. This possibility was investigated in a pilot project. These 41 ISGs were examined in small subsets of PBMC (*n* = 4, HESN and *n* = 5 non-HESN), then subjected to a 3-h stimulation by exogenous interferon-*γ* (IFN-γ, 10 ng/mL) and IFN-α2 (10 ng/mL). Similar increases in the transcripts of 36 ISGs (out of 41) were found in both HESN and non-HESN groups (data not shown, Table 2: bolded ISGs). Among the 36 responsive ISGs, 8 ISGs (*CASP8, FLIP, GADD45, IRF4, IRF5, MxB, p53, TAP1*) showed trending differences between the 2 groups (12–38% difference in transcript levels, *p*-values ranged 0.065 to 0.09). Five ISGs remained undetected following stimulation with IFN-γ and IFN-α2 (Table 2: underlined ISGs). These 41 ISGs did not seem to be differentially regulated in HESN PMBCs.

The study focused on profiling the 35 ISGs listed in Table 3. Transcripts of all 35 ISGs were detected in the PBMC from non-HESN control (*n* = 26) but only 25 ISG transcripts (at varying concentrations) were detectable in the HESN PBMC (*n* = 23), suggesting potentially reduced basal expression of these 10 ISGs in the HESN FSWs (Table 3). When examining the properties of the 10 ISGs that were constitutively expressed only in the non-HESN controls, these ISGs were enriched in the signaling pathways of apoptosis, and pathogenic hyper-cytokinemia/-chemokinemia (ingenuity pathway analysis, *p*-values <10^−5^), potentiating a more proinflammatory milieu in the non-HESN FSWs. Contrarily, HESN PBMCs had less expression of most ISGs with antiviral properties, supporting a less inflammatory microenvironment.

Among the 25 ISGs that were constitutively expressed in the PBMC of both HESN and non-HESN controls, 11 ISGs had differential expression levels between the two groups (Table 3, bolded), while the other 14 ISGs were equivalent. The transcript levels of *CXCL10, IRF-1, IFIT1, IFNB1, MX1, COX2*, and *TNF* were significantly lower by 61–94% in the PBMC of the HESN FSWs, compared to that in the non-HESN controls (i.e., 1.5- to 16.6-fold more transcripts in the non-HESN controls) (Figure 1). These ISGs were enriched in the pathways of antiviral response and the replication of virus RNA in immune cells (ingenuity pathway analysis, *p*-values < 10^−^^5^). Reduced expression of these ISGs reflected a weak basal antiviral readiness in the HESN PBMC. However, the transcripts of two key signaling molecules of proinflammatory antiviral response were significantly higher in HESN PBMC. The levels of *IFIT3* and *IFIH1/MDA5* transcripts were 2.3-fold (*p* = 0.04) and 11.3-fold (*p* = 0.003) higher in HESN PBMC. Both *IFIT3* and *MDA5* have RNA-binding properties [67,68,69], and are involved in responses against DNA and RNA virus infection, suggesting that HESN PBMCs are more ready to detect RNA or DNA viral infection.

### 3.2. The PBMC of HESN FSWs Had Reduced Transcript Levels of Antiapoptotic ISGs and Elevated Proapoptotic TRAILR1 Transcripts

In addition to antiviral function, some ISGs also play a crucial role in regulating cellular apoptosis [45,47,65]. The HESN PBMCs had no detectable level of *BCL2* or *BCL2L.1* transcripts, encoding crucial anti-apoptotic molecules at baseline (Table 3). In addition, the HESN PBMCs had a significantly higher RNA transcript level of *TNFRSF10A/TRAILR1*, a critical inducer of apoptosis, compared to the HESN controls (by 26-fold, *p* < 4 × 10^−6^, Figure 1). Although it seems that the HESN PBMCs have greater pro-apoptotic potential, the transcript level of *IFI6*, an anti-apoptotic regulator and immune modulator was doubled in the HESN PBMC, versus the non-HESN controls (*p* =0.01, Figure 1), suggesting a mechanism of balance.

### 3.3. IFN-γ Induced the Expression of Pro-Inflammatory ISGs in HESN PBMC, Although to a Lesser Extent than in the Non-HESN PBMC, but Greatly Enhanced the Expression of Pro-Apoptosis ISGs

The transcriptional response of the 35 detectable ISGs to exogenous IFN-γ treatment was evaluated in HESN and non-HESN PBMCs. Not all 35 ISGs responded following the 3-h stimulation with exogenous IFN-γ; 19 ISGs had only a modest 12–25% increase or decrease in transcript level in either the HESN or the non-HESN PBMCs (data not shown). Among these 19 ISGs, 7 ISGs (*ACTB, E2F1, FCGR2B, IL-1B, PKR, MYD88, TNF*) did not mount a statistically significant response in either group when compared to the mock stimulation (*p* > 0.05). IFN-γ induced changes in the other 12 ISG transcript levels (*ATM, BAX, BclXL, CDK4, CDKN2A, DPP4, HDM2, IFI6, IFIH1, IFNA2, ISG15, NOXA*) were similar between the HESN and non-HESN PBMCs (*p*-values > 0.05, data not shown).

The other 16 ISGs responded with either a >4-fold increase or a >25% decrease in transcripts in either the HESN PBMCs or the non-HESN controls (Figure 2). IFN-γ up regulated the expression of 11 ISGs (*CXCL10, IRF-7, IFNB1, IRF-1, RANTES, MX1, COX2, IRF-9, IFIT3, STAT1A, and TRAIL*) in both the HESN and non-HESN PBMC samples, ranging from 113-fold to 4-fold (Figure 2, HESN, *p*-values <0.0001; non-HESN, *p*-values range from 0.0001 to 0.02). The increases in the transcript levels of *CXCL10, IRF-7,* and *IFNB1* were similar in both groups (*p*-values >0.05), while the increases in *IRF-1* (31-fold in non-HESN vs. 4-fold in HESN), *RANTES* (10- vs. 6-fold), *MX1* (22- vs. 5-fold), *COX2* (24- vs. 7-fold), *IRF-9* (11- vs. 7-fold), *STAT1A* (12- vs. 7-fold), and *TRAIL* (16- vs. 4-fold) were notably greater in the non-HESN PBMCs. The increase in *IFIT3* transcripts, encoding for a sensor of RNA virus was much higher (*p* < 0.0001) in the HESN PBMC (by 34-fold) versus that in the non-HESN controls (by 5-fold), boosting the antiviral potential in HESN. In addition, IFN-γ up regulated *ISG56* (4-fold), *IFNA4* (10-fold), and *RIG-1* (6-fold) transcripts only in the non-HESN PBMCs, without affecting the expression of *ISG56, IFNA4*, and *RIG-1* in the HESN PBMCs (Figure 2), suggesting a less pro-inflammatory response in HESN. Together, IFN-γ stimulation seemed to further enhance the differences in the baseline ISG transcript levels between the HESN and the non-HESN controls (Table 3: bolded ISGs and non-HESN only ISGs), showing reduced inflammatory potential in HESN PBMCs when responding to IFN-γ but enhanced antiviral readiness via increased *IFIT3*.

Similarly, IFN-γ seemed to enhance expression of pro-apoptotic potential in HESN PBMC, while boosting anti-apoptotic transcripts in non-HESN PBMCs. IFN-γ stimulation down regulated the expression of anti-apoptotic ISG, *BCL2* by 74%, (vs. mock-stimulation, *p* = 0.002) and up-regulated the initiator of the apoptotic pathway, *TRAILR1*, by 10-fold (vs. mock-stimulation, *p* < 0.0001) in the HESN PBMCs. IFN-γ stimulation almost had the opposite effects on the ISGs regulating apoptosis in the non-HESN PBMC. IFN-γ down regulated *TRAILR1* expression by 38%, (vs. mock-stimulation, *p* =0.02) but had no effect on the *BCL2* expression (*p* > 0.05) in the non-HESN PBMCs, reducing the overall apoptotic potential of the non-HESN PBMC. It is noteworthy that IFN-γ up regulated *TRAIL* expression in both groups, supporting the initiation of apoptosis but did so with a significantly greater impact on the non-HESN PBMC (13-fold increases, *p* < 0.0001) and a relatively less propensity for apoptosis for the HESN PBMC (4-fold, Figure 2).

### 3.4. The Transcriptional Response of Antiviral Effector ISGs Was Transient in the HESN PBMC and CD4+ T Cells, Compared to the Prolonged Response in the Non-HESN Controls

A prolonged state of immune activation or inflammation and/or resistance to apoptosis that is induced by interferons or viral infection has been associated with pathogenic outcomes [51,52,70,71]. The kinetics of ISG transcriptional responses to IFN-γ were studied at 1 h, 3 h, and 16 h following the initial exposure to exogenous IFN-γ (10 ng/mL) to determine the duration of ISG response. The antiviral effector ISGs, including *IRF-1, CXCL10, IFNB1, IRF-9*, and *IFIT3* were examined to profile the changes of ISG transcripts over time in both groups (Figure 3). *IRF-1* was included as a positive control, as our earlier work showed transient *IRF-1* transcript increases in the HESN [41,43]. Expression of *CXCL10* and *IFNB1* can be regulated by IRF-1 [56,57,58] while expression of *IRF-9* and *IFIT3/ISG60* is not dependent on IRF-1. Similar to what we observed for *IRF-1*, the transcript level of the other antiviral effector ISGs increased drastically within an hour in both the HESN and non-HESN PBMCs (Figure 3A). However, 3-h post IFN-γ stimulation, transcripts levels dropped in HESN PBMC but continued to increase or plateau in the non-HESN PBMCs. At 16 h post IFN-γ stimulation, ISG transcripts in the HESN PBMC returned to the baseline level while ISG transcripts in the non-HESN PBMC remained elevated. Significant differences in the ISG transcript levels of *IFNB1, CXCL10, IRF-1*, and *IRF-9* were observed between the two groups at 16 h (Figure 3A). Interestingly, the *IFIT3* transcript level in non-HESN PBMC plateaued at the 16th hour poststimulation, reaching the same level as the *IFIT3* transcripts in the HESN PBMC, suggesting that, when activated, *IFIT3* responds with a strong transient ISG upregulation, rather than a sustained, slow response pattern.

While assessing ISG expression and response in PBMC enabled the evaluation of systematic ISG regulation, representing a general antiviral response, assessing ISG regulation in the CD4+ T cells allowed the evaluation of immune activation of the main target for HIV infections. This is of importance because HIV-1 replicates in activated CD4+ T cells [72,73]. The study determined whether ISG responses to IFN-γ in the CD4+ T cells followed similar kinetics as to bulk PBMC (Figure 3B). Similar patterns of ISGs responses were observed in CD4+ T cells in both HESN and non-HESN groups. Surprisingly, elevated *IRF-9* transcripts in the HESN CD4+ T cells did not return to the baseline level, as in the HESN PBMC. However, at 16 h poststimulation, the *IRF-9* transcript level in the HESN CD4+ T cells was still lower than that in the non-HESN CD4+ T cells, suggesting a slightly prolonged but controlled IRF-9 mediated activation of IFN-responsive genes in the CD4+ T cells, compared to other cell types in the PBMC.

### 3.5. IFN-γ Transiently Upregulated Proapoptotic Pathway ISG (TRAILR1 and TRAIL) and Downregulated Antiapoptotic ISG, BCL2 in HESN CD4+ T Cells

We also characterized the kinetics of the transcriptional responses of two ISGs encoding proteins regulating cellular apoptosis, *TRAILR1* and *BCL2* in the PBMC and CD4+ T cells of both the HESN and the non-HESN FSWs (Figure 4). The baseline expression of these two ISGs were significantly different in the HESN and non-HESN controls (Figure 1 and Figure 4). Following the IFN-γ challenge, the transcripts of the proapoptotic receptor gene, *TRAILR1* were increased within 1 h in PBMC (HESN: 31-fold vs. non-HESN 3-fold) and the CD4+ T cells (HESN: 5.5-fold vs. non-HESN 2.7-fold) of both the HESN and the non-HESN controls. The *TRAILR1* transcript level was quickly reduced at 3-h poststimulation in the PBMC and CD4+ T cells of both groups. At the 16th hour poststimulation, the *TRAILR1* transcript returned to the baseline level in the PBMC and CD4+ T cells of both groups. Although the kinetics were similar for the *TRAILR1* in both the HESN and the non-HESN controls, the magnitude of changes in the *TRAILR1* transcripts was significantly larger in the HESN group, suggesting a very tight regulation of *TRAILR1* expression in the HESN cells (Figure 4).

In contrast to its effects on *TRAILR1*, IFN-γ reduced the level of *BCL2* transcripts, an anti-apoptotic/pro-survival ISG in the PBMC of both the HESN (by 64%) and the non-HESN controls (by 27%) within the first hour of stimulation. *BCL2* transcript levels remained low in the PBMC of both groups (Figure 4), up to 16 h post-IFN-γ exposure. The effects of IFN-γ on *BCL2* transcripts in the CD4+ T cells, however, did not mirror those in the PBMC. Similar reduction of *BCL2* transcripts in HESN CD4+ T cells were observed during the first 3 h of IFN-γ challenge, but at 16 h poststimulation, the *BCL2* transcripts returned to baseline level, instead of remaining reduced, as observed in the HESN PBMC. It is not clear whether there is a functional gain for restoring *BCL2* transcript level in CD4+ T cells and whether that contributes to the HESN phenotype.

Furthermore, IFN-γ treatment of the non-HESN CD4+ T cells increased *BCL2* transcripts (2.7-fold); together with the reduced *TRAILR1* expression, IFN-γ protected against apoptosis in the non-HESN CD4+ T cells. In contrast, IFN-γ stimulation transiently primed the HESN CD4+ T cells to initiate apoptosis with increased *TRAILR1* and decreased *BCL2* transcripts.

### 3.6. IFN-γ Induced a Stronger Negative Feedback Regulatory Response in the HESN PBMC, When Compared to Non-HESN Controls

SOCS proteins have been shown to be induced by interferons as part of a negative feedback loop to limit the extent and duration of IFN response [74]. This study examined whether there is a differential regulation of suppressor of cytokine signaling-1 (SOCS1) expression in the HESN versus the non-HESN controls. (Figure 5). Basal level of *SOCS1* transcripts in HESN PBMC was not different from the *SOCS1* transcript level in the non-HESN PBMC (Figure 5A). At 3-h following IFN-γ stimulation, the *SOCS1* transcript levels were significantly upregulated in both the HESN and the non-HESN PBMC (Figure 5B). Furthermore, in comparing the extent of *SOCS1* upregulation in the HESN cells with that in the non-HESN cells, *SOCS1* transcripts were increased by approximately 16-fold in the HESN PBMC, versus a 4-fold increase in the non-HESN PBMC (*p*-value = 0.002, Figure 5C). Together, these data showed that IFN-γ induced a much stronger negative feedback signal in the HESN PBMCs, when compared to the non-HESN controls, suggesting a tighter control of the IFN-γ induced ISGs in the HESN.

### 3.7. The Transcriptional Response of Antiviral Effector ISGs and Antiapoptotic ISGs to Exogenous IFN-α2 Stimulation Was Similarly Transient in HESN PBMC and CD4+ T Cells

Similar to IFN-γ, the type-I IFN, IFN-α, is also a strong inducer of antiviral ISGs [49,60,66,75] and uncontrolled IFN-α responses have been associated with increased pathogenesis [51,52,70,71]. This study examined whether the kinetics of ISG responses to IFN-α treatment differed from that by IFN-γ in PBMCs (Figure 6). IFN-α2, the prototypical IFN-α and a key part of the innate immune response with potent antiviral, antiproliferative, and immunomodulatory properties was used. The expression of *IRF-1, IRF-9/ISGF3G, FcGR1A*, and *HDM2* transcripts dramatically increased within an hour following IFN-α2-treatment in the PBMC of both the HESN and the non-HESN controls (Figure 6(Ai–Aiii,Av,Avi), *p*-values < 0.0001). Similar to the results of IFN-γ stimulation, a reduction in *IRF-1, IRF-7*, and *IRF-9* transcript levels was observed in HESN PBMC at 3-h post exposure to IFN-α2; The transcript levels of *IRF-1, IRF-7*, and *IRF-9* seemed to plateau in non-HESN PBMC at the 16th hour post-treatment (Figure 6(Ai–Aiii)). Unlike the responses to IFN-γ, the *IRF-9* transcripts only started to decrease at 16-h post-treatment (*p* = 0.03) and the *IRF-1* and *IRF-9* transcripts were not restored to the baseline level in the HESN PBMCs. Overall, HESN PBMC responded to IFN-α2 stimulation with transient increases in the antiviral effector ISGs (Figure 6(Ai–Aiii)). These transient increases of *IRF-1, IRF-7*, and *IRF-9* were also observed in the HESN CD4+ T-cell populations (Figure 6(Bi–Biii)). Interestingly, the maximum *IRF-7* transactivation response to IFN-α2 in the HESN PBMC (64-fold (HESN) vs. 4-fold (non-HESN)) or CD4+ T cells (142-fold (HESN) vs. 18-fold (non-HESN)) were much more robust than that in the non-HESN cells, suggesting a potentially strong but transient inflammatory response to IFN-α2 in the HESN CD4+ T-cell populations.

When we examined the pro-apoptotic marker *BCL2*, similar to the kinetics of *BCL2* response, elicited by IFN-γ (Figure 4), *BCL2* transcript levels dropped in the PBMC of both HESN and non-HESN during the first 3 h of IFN-α2 treatment (Figure 6(Aiv)). However, instead of remaining at low levels, as observed during IFN-γ stimulation (Figure 4), the *BCL2* transcript level began to increase at the 16th hour post-treatment in the PBMC of both HESN and non-HESN (Figure 6(Aiv)). In CD4+ T cells, the kinetics of *BCL2* response to the IFN-α2 (Figure 6(Aiv)) was the same as the *BCL2* response to IFN-γ (Figure 4). In HESN CD4+ T cells, the drop of *BCL2* transcripts was transient (by 38–84%) compared to non-HESN, where IFN-α2 treatment increased the *BCL2* transcripts by 1–2 fold (Figure 6(Biv)). This suggests regulation of apoptosis in the CD4+ T cells of HESN, is altered compared to non-HESN. By dampening the pro-survival ISG, *BCL2* expression to allow apoptosis was quite exact during both IFN-γ and IFN-α2 stimulation.

### 3.8. Not All Antiviral ISGs in the HESN Cells Exhibited Transient Response to IFN-α2 Treatment

*ISG15* is a ubiquitin-like protein that becomes conjugated to many cellular proteins upon activation by type-I IFNs [76], while *HDM2* gene product is critical for the negative feedback of ISG activation via dampening the expression and function of IRFs in antiviral response [77,78]. The baseline level of *ISG15* and *HMD2* transcripts were very low in both the PBMC and CD4+ T cells of both groups (Figure 6(Av,Avi,Bv,Bvi)); *ISG15* transcripts were not detected in HESN cells at baseline (Table 3, Figure 6(Av,Bv)). The relative level of transcripts detected were close to the water control blank in the RT-qPCR (ct-value < 37). However, following stimulation with IFNα2, *ISG15* and *HMD2* transcripts increased steadily in both HESN PBMC and CD4+ T cells, and reached plateaus in the non-HESN PBMCs. The fact that not all ISGs appear to be alternatively regulated in HESN suggests a specific pattern of ISG expression may be associated with protection from HIV infection. Steady increases in the expression of these two ISGs might be critical for regulating antiviral response.

### 3.9. Reducing IRF-1 Expression in Non-HESN CD4+ Cells Altered the Non-HESN ISG Profile to the ISG Profile, Similar to What Was Observed in the HESN Cells

This next step was to determine if a moderate reduction of baseline IRF-1 in isolated CD4+ T cells, like that observed in the HESN FSWs (by 30–60%), would mimic the HESN ISGs profile. IRF-1 expression was knocked down with IRF-1 specific siRNA, delivered by CD4-specific RNA aptamer chimera (CD4-AsiCs, *n* = 6 per group). CD4-AsiCs with scrambled siRNA were used as negative control. Greater than 95% of CD4+ T cells were shown to have IRF-1 protein level reduced by 30–60% (data not shown, [41]). Due to the limitation in cell numbers, only the expression of key antiviral ISGs shown to be regulated by IRF-1 were examined at 18 h (Figure 7). The baseline level of *IFN-γ, TNF-α, STAT1α, RIG-1*, and *IFIH1* transcripts were not affected by reduced *IRF-1* expression, suggesting that the basal expression of these genes is independent of the *IRF-1* expression level. *IRF-1*-specific CD4-AsiC reduced cellular *IRF-1* transcripts by 99.7%, reflecting a 30–60% decrease in IRF-1 protein level at 18-h. The transcript levels of two proinflammatory *ISGs, MX1* (by 96%) and *IRF-7* (by 88%) were greatly reduced (Figure 7) while transcripts of the RNA-sensing *ISG, IFIT3*, were increased by 283-fold. The resulting *MX1, IRF-7,* and *IFIT3* transcript levels were similar to the low *MX1* (Figure 1) and *IRF-7* (Figure 6(Biii)) and high *IFIT3* (Figure 3B) levels observed in HESN (solid circles). In addition, the resulting transcript levels of the proapoptotic ISG, *TRAILR1*, and antiapoptotic regulator, *IFI6*, were increased by 22-fold and 62-fold, respectively (Figure 7), similar to the *TRAILR1* and *IFI6* transcript levels observed in HESN (Figure 1 and Figure 4). Together, reducing *IRF-1* expression in non-HESN CD4+ T cells altered the transcript levels of *MX1, IRF-7, TRAILR1, IFIT3*, and *IFI6* to similar levels found in HESN. A high IRF-1 level seemed to negatively regulate *IFIT3, TRAILR1*, and *IFI6*, but positively promote the expression of *MX1* and *IFIH1*.

## 4. Discussion

Antiviral ISGs are a critical response to viral infection; in vitro loss of function studies showed that knockout of antiviral ISGs resulted in increased replication of several viruses [48,49,59,66,75]. This is the first study exploring ISG expression as a correlate of protection against HIV-1 infection. This study identified a unique profile of antiviral ISGs in HESN FSWs that is characterized by (1) a lower level of ISG transcripts that potentiate a proinflammatory response, (2) a balance of proapoptotic ISGs and prosurvival ISGs, and (3) transient increases of proinflammatory ISGs and ISGs with apoptotic potential. This study further showed that reduced *IRF-1* expression, observed in the HESN cells may be one of the key regulators of this HESN ISG profile.

Although increased inflammation has been associated with increased risk to HIV infection [79,80,81,82,83,84] and baseline immune quiescence (IQ) was observed in these Kenyan HESN FSWs [26], little is known of the mechanism(s) involved in the maintenance of IQ. IQ refers to a state of low baseline immune activation, which was proposed to protect against infection by limiting activated HIV target cells [26,62] and was associated with increased regulatory T cells [38]. Baseline IQ is not to be mistaken as “immune deficiency”. Even though the baseline level of several antiviral ISG transcripts in the HESN cells were significantly less than that in the non-HESN cells, the two groups mounted equivalent magnitudes of ISG responses to exogenous IFNs (Figure 2, Figure 3 and Figure 6). This is especially important because IFN is one of the primary innate immune responses against viral infection [51,52,59]. HIV-1 infection elicits type-I, type-II and type-III IFN responses, which activate the transcription of ISGs [46].

Uniquely to the HESN cells, the proinflammatory and proapoptotic ISG-transcription response was transient and appeared to be promptly downregulated after 3 h. In comparison, the ISG mRNA transcript level continued to increase or reach a plateau in the non-HESN cells, which may result in prolonged inflammation. Activation of immune cells, especially CD4+ T cells, is essential for HIV-1 replication and the establishment of infection [80,85,86,87]. Previous studies suggest that sustained elevation of type-I IFN expression is associated with persistent immune activation or inflammation, which promotes HIV-1 replication and transmission. The persistently high expression of ISGs has been shown to be a major contributor of inflammation and the resultant establishment of infection [51,52,59] and disease progression during persistent HIV-1 and SIV infections [51,52,70,88,89,90]. Hence, the regulation of these ISGs, as observed in HESN cells, may be critical for effective antiviral response with minimum pathogenesis.

Negative regulators of immune activation are critical for resolving the IFN-induced antiviral state and facilitating the return to immune homeostasis. In addition to reduced *IRF-1* expression, this study found rapid upregulation of *SOCS-1*, a negative regulator of IFN-signaling in HESN individuals. The HESN cells had four-times-higher levels of *SOCS-1* than the non-HESN cells, following IFN-γ stimulation. SOCS-1 has also been shown to negatively regulate *IRF-1* expression [91]. While the baseline level of *SOCS-1* transcripts, which are similar in both the HESN and the non-HESN groups could not explain the reduced *IRF-1* basal expression, the increased *SOCS-1* transcripts following IFN-γ stimulation may contribute to the transient *IRF-1* response. Furthermore, *SOCS-1* polymorphisms have been associated with a rapid HIV progression rate [92]. Recently, in in vivo genome-wide CRISPR screens, *SOCS1* was identified as “a major non-redundant checkpoint imposing a brake on CD4+ T-cell proliferation,” which promotes HIV-1 replication in infected cells [93]. This is in agreement with the brief, robust, but transient ISG response observed in the HESN cells, suggesting that SOCS-1 may be another key molecule enabling the HESN phenotype.

The regulatory mechanism(s) underlying reduced baseline *IRF-1* expression and increased *SOCS-1* expression following IFN-γ stimulation in the HESN cells need further research. Thus far, our earlier study showed that the IRF-1 genetic polymorphisms, found in approximately 72% of the HESN FSWs are associated with reduced *IRF-1* expression, delayed *IRF-1* response to IFN-γ and delayed establishment of HIV-1 infection [14]. However, the *IRF-1* polymorphism alone could not explain the reduced *IRF-1* expression in the HESN cells. The HESN FSWs without the protective *IRF-1* polymorphism also exhibited reduced baseline *IRF-1* levels. Our work suggested that epigenetic mechanism(s), which are influenced by environment and diets, may contribute to the regulation of *IRF-1*. On the note of environmental influences, the HESN FSWs were in the sex trade for a significantly longer period of time and had been exposed to more clients; it is highly possible that the HESN phenotype was driven by high exposure to foreign antigens, which induced immune tolerance to avoid inflammation at the mucosal site [94]. However, the mechanism(s) of such remained to be sought. IRF-1 regulates the differentiation of Treg, via suppressing *FOXP3* transactivation [95]; reduced IRF-1 expression, perhaps affected by levels of metabolites (e.g., short-chain fatty acids) may induce immune tolerance via increasing differentiation of Treg.

Interestingly, while SOCS1 can regulate *IRF-1* [91], *SOCS1* was shown to be regulated by IRF-1 as well [74,96]. The *SOCS1* promoter consists of IRF element (IRF-E); IRF-1 binding to IRF-E is necessary to transactivate the expression of *SOCS1* [96]. This study highlights the importance to define the inter-regulatory relationship between IRF-1 and SOCS1 to gain a better understanding of the HESN phenotype.

In addition to inflammation, an increased potential in eliciting an apoptotic pathway may also contribute to the HESN phenotype. Programmed cell death (i.e., apoptosis) of infected cells is critical to limit the spread of pathogens and inappropriate immune activation [45,47,66,97]. However, when the surrounding tissue is also severely damaged, via bystander effects, the damaged tissue would enhance the proinflammatory response and disease severity is often augmented. The regulation of pro-apoptotic and pro-survival ISGs was found to be unique in the HESN cells. IFN stimulation induced proapoptotic ISGs (*TRAILR1, TRAIL*) but reduced the expression of pro-survival ISG, *BCL2* (Figure 2 and Figure 4). *TRAIL* (TNF-related apoptosis-inducing ligand) is induced in several cell types including immune cells such as macrophages and T cells [47,59] and its expression has been associated with induction of apoptosis of the *TRAIL*-expressing cells and the surrounding bystander cells and further shown to affect disease outcomes. This study found that HESN cells had a 4-fold increase in *TRAIL* mRNA transcripts following IFN-γ stimulation, compared to a 16-fold increase observed in the non-HESN cells (Figure 3). Contrary to that, the HESN cells had a 15-fold increase in the expression of TRAIL-receptor, *TRAILR1* but the non-HESN cells have no significant change in *TRAILR1* expression following IFN-γ challenge. Together, these data suggest a proapoptotic milieu promoting cell death of bystander cells in non-HESN; but in HESN individuals, perhaps a transient proapoptotic state may be more protective against immune activation, and bystander cell/tissue death and damage. In support, the pro-survival ISG, *BCL2* was also transiently reduced in the HESN cells. The HESN cells seem to have a strict, fine control of these pro- and anti-apoptotic ISGs to quickly maximize the death of affected cells and minimize damages to surrounding cells.

Lastly, our earlier work and findings from this study support IRF-1 as playing a key role in the maintenance of IQ in HESN FSWs. IRF-1 is not only a trans-activator of several pro-inflammatory ISGs; it is also a repressor of *FOXP3* and, thus, an inhibitor of the differentiation of regulatory T cells (Treg). With reduced IRF-1 protein (30–60%), as found in the HESN FSWs [42,98], the expression of *IRF-1*-dependent ISGs were decreased (Figure 1 and Figure 7), and the expression of *FOXP3* and *TRAILR1* were increased, perhaps leading to increased Treg differentiation and apoptotic potential, respectively. IRF-1′s crucial role in HIV-1 susceptibility was further supported by its upregulation by HIV-1 infection and the consequence of knocking down its expression using siRNA. IRF-1 also binds to the HIV-1 promoter; together with NF-κB, IRF-1 is required for the trans-activation of viral genes. *IRF-1* knockdown resulted in significant reduction and delay of HIV-1 replication [41]. Together, these support the notion that reduced baseline *IRF-1* expression is key to avoiding the establishment of HIV infection in HESN.

Limitations/caveats: There are several limitations in the design of the study. As this was a retrospective analysis, we were unable to select large numbers of cryopreserved samples to best fit the grouping criteria due to limitations of the bio-repository. In addition, we had to limit analysis to a limited number of cell samples with the viability >90% after thawing. Furthermore, as mentioned in the methodology, the definition criteria for HESN and non-HESN was based on the epidemiological data. To control for the demonstrable effects of sex work on immune activation, we used FSWs who have not been in the sex trade long enough to meet the HESN definition. Over the years of follow-up, there were significant numbers of seroconversions among non-HESN FSWs and there was loss to follow-up as several of the participants did not return for the final resurvey in this cohort, which ended in 2013. Furthermore, this study focused on a subset of antiviral ISGs that have been implicated in several studies of innate responses to viral infection [45,47,56,58,65,66], instead of unbiased analysis of all known ISGs. There are more than a thousand ISGs; hence, it is possible that several potential ISGs that are critical for the HESN phenotype were omitted in this study and that data from this study is only a tip of an iceberg. On a separate study, RNA-seq was performed with four PBMC samples from each of the HESN and the non-HESN group (data not shown); only a few ISG were found to be differentially expressed between the two groups, and most of those have been included in this study. This study also focused on CD4+ T cells; future study should focus on RNA from different cell subsets. The expression of ISGs is tightly regulated and specific to cell types; hence, levels of ISG transcripts may be diluted in the RNA pool from PBMC. This study avoided using ISG super-arrays and RNA-seq to include more samples in the analyses due to sample limitation and to include the kinetic study and to allow presentation of validated data, as only validated primer sets were used. The data is interpreted with caution so as not to emphasize that these ISGs are the only molecules contributing to the HESN phenotype. Lastly, although there are significant discrepancies between the cells in the blood and the cells found at the mucosal compartments [99], findings from studies of PBMCs are highly valuable in guiding research at the vaginal site.

## 5. Conclusions

The regulation of IFN signaling and ISG expression are unique in HESN cells and may be part of the molecular mechanisms underlying the reduced susceptibility of HESN FSWs to HIV-1 infection. The transient ISG response observed in the HESN cells may be mediated by the drastic increases of SOCS1, short-lived upregulation of IRF-1 and reinforced by increasing HDM2 and subsequent regulation. Together with the coordinated regulation of proapoptotic and prosurvival ISGs, IRF-1-regulated ISGs founded the basis for the HESN phenotype observed in the Kenyan FSWs.

## Figures and Tables

**Figure 1 viruses-14-00471-f001:**
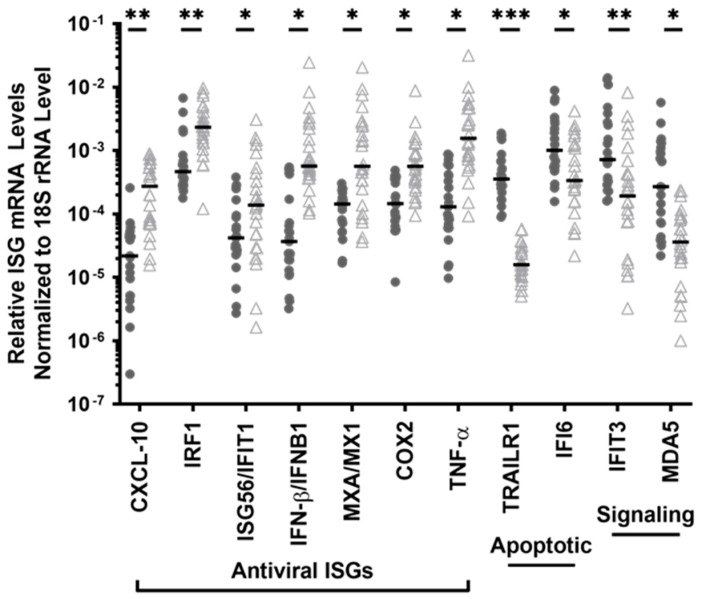
Basal expression of 11 ISGs in the ex vivo PBMC differed between the HESN (•) and the non-HESN controls (△). Cellular RNA transcript levels were quantitated using RT-PCR, and total transcript levels were normalized to 18S rRNA. Median values (**-**) were shown for each group. Multiple *t*-tests between the HESN and non-HESN groups were performed (Prism 9.0); *n* = 23 HESN, and *n* = 26 non-HESN controls. * *p*-value < 0.05; ** *p*-value < 0.01, *** *p*-value < 0.001.

**Figure 2 viruses-14-00471-f002:**
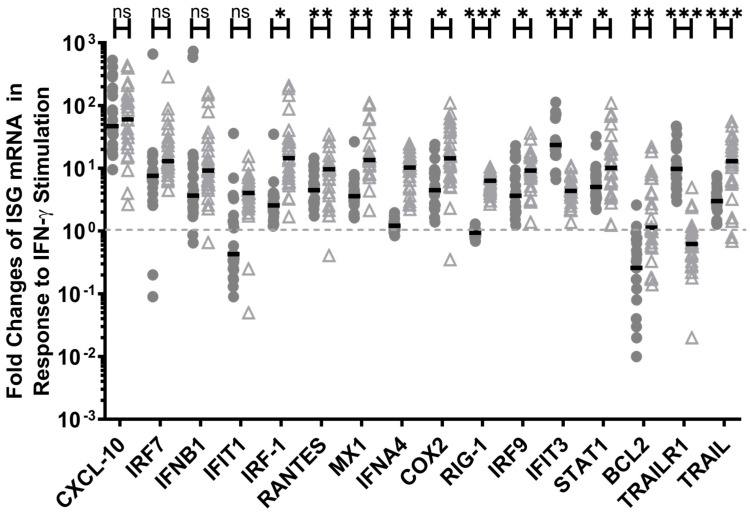
Changes in a subset of antiviral ISG gene expression in ex vivo PBMC, in response to exogenous IFN-γ stimulation. Three hours post-stimulation, cellular RNA transcript levels were quantitated using RT-qPCR, and total transcript levels were normalized to 18S rRNA. Multiple *t*-tests between the HESN (•) and the non-HESN (△) groups were performed (Prism 9.0); *n* = 23 HESN, and *n* = 26 non-HESN controls. Median values (**-**) were shown for each group. * *p*-value < 0.05; ** *p*-value < 0.01, *** *p*-value < 0.001; n.s. denotes ‘not statistically significant’.

**Figure 3 viruses-14-00471-f003:**
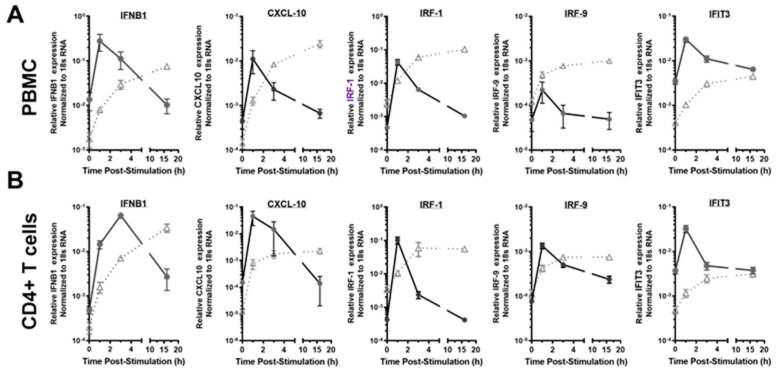
Kinetics of transcriptional response to exogenous IFN-γ stimulation in PBMC and CD4+ T cells. (**A**) PBMC and (**B**) enriched CD4+ T cells from the HESN (• *n* = 12) and the non-HESN (△, *n* = 10) study participants were stimulated with IFN-γ (10 ng/mL) for the indicated time (hour, h). Levels of nascent transcripts of antiviral effector ISGs, including *IRF-1, CXCL-10, IFNB1, IRF-9, and IFIT3* were assessed using RT-qPCR and normalized to 18S rRNA levels. We performed *t*-tests or the transcript levels at each time point between the two study groups.

**Figure 4 viruses-14-00471-f004:**
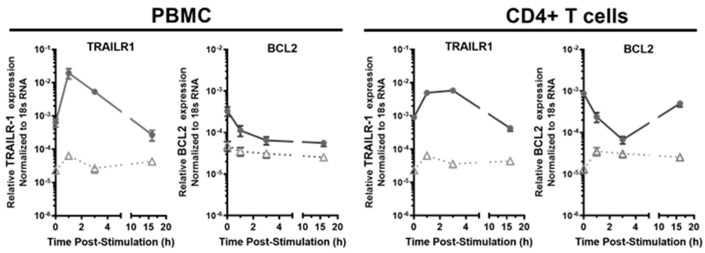
The kinetics of the transcription of apoptotic genes, TRAILR-1 and BCL2 in PBMC and CD4+ T cells in response to exogenous IFN-γ. PBMC and enriched CD4+ T cells from the HESN (• *n* = 12) and the non-HESN (△, *n* = 10) study participants were stimulated with IFN-γ (10 ng/mL). Total RNA was isolated at the indicated time, following stimulation. Levels of nascent transcripts were assessed using RT-qPCR and normalized to 18S rRNA levels.

**Figure 5 viruses-14-00471-f005:**
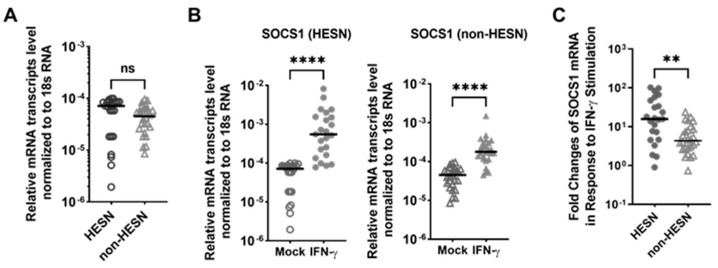
Expression of *SOCS1* mRNA in HESN and non-HESN PBMC. Relative levels of *SOCS1* RNA transcripts in HESN and non-HESN PBMC were evaluated (**A**) before and (**B**) 3 h after IFN-γ stimulation (10 ng/mL). Levels of *SOCS1* transcripts were normalized to 18S rRNA levels. (**C**) Fold changes in *SOCS1* transcript levels, included by IFN-γ were calculated. *t*-tests were used to compare the differences between the two groups. Median values (**-**) were shown for each group. ** *p*-value < 0.01, **** *p*-value < 0.0001; n.s. denotes ‘not statistically significant’.

**Figure 6 viruses-14-00471-f006:**
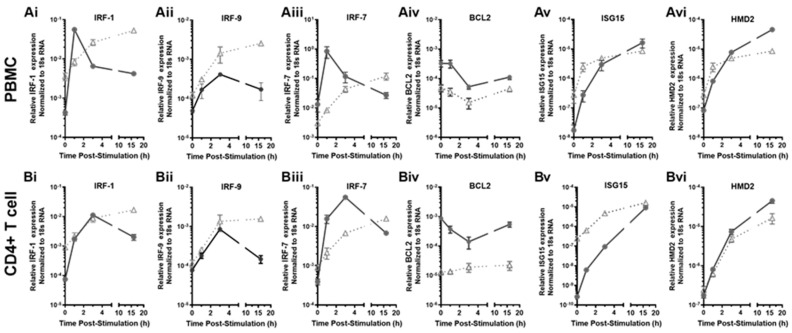
The kinetics of the transcription of ISGs in PBMC and CD4+ T cells in response to exogenous IFN-a2. (**A** panel) PBMC and (**B** panel) enriched CD4+ T cells from the HESN (• *n* = 12) and the non-HESN (△, *n* = 10) study participants were stimulated with IFN-a2 (10 ng/mL). Total RNA was isolated at the indicated time, following stimulation. Levels of nascent transcripts of (**Ai**,**Bi**) IRF-1, (**Aii**,**Bii**) IRF-9, (**Aiii**,**Biii**) IRF-7, (**Aiv**,**Biv**) BCL2, (**Av**,**Bv**) ISG15, and (**Avi**,**Bvi**) HMD2 were assessed using RT-qPCR and normalized to 18S rRNA levels.

**Figure 7 viruses-14-00471-f007:**
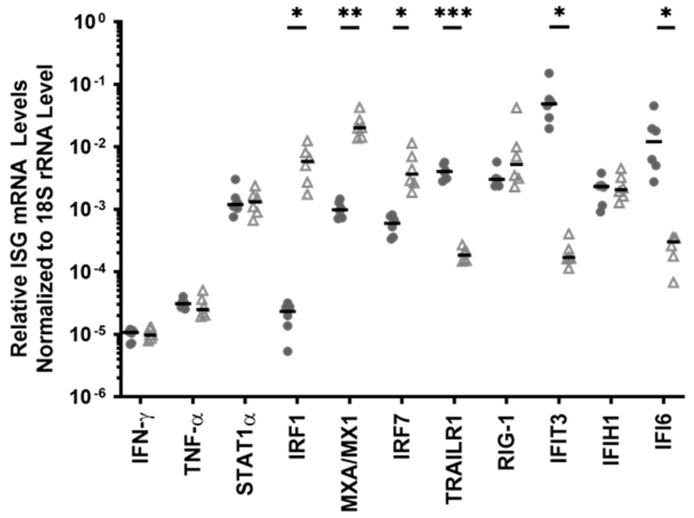
A modest reduction in *IRF-1* expression altered the ISG expression profile in unstimulated primary CD4+ T cells from the non-HESN controls. Ex vivo CD4+ T cells were treated with *IRF-1* specific (•), or control siRNA (△) aptamer chimera (8 nM) that enters cells via binding to cell surface CD4 molecule. At 24 h, the efficiency of RNA knockdown was assessed using flow cytometry and ISG RNA transcript levels in the CD4+ T cells were quantitated using RT-qPCR. All relative expression values were normalized to the corresponding 18S rRNA. (*n* = 6 per group). Median values (**-**) were shown for each group. * *p*-value < 0.05; ** *p*-value < 0.01, *** *p*-value < 0.001; n.s. denotes ‘not statistically significant’.

**Table 1 viruses-14-00471-t001:** Clinical characterization of study subjects at enrolment.

Parameter	HESN (*n* = 23)	Non-HESN (*n* = 26)	*p*-Value ^a^
	Median or Number (Range or %)		
Age (years)	37 (26–45)	31 (21–38)	<0.001
Weight (Kg)	77 (52–98)	71 (42–96)	n.s.
Active in sex work (years)	12 (7–24)	2 (0–3)	<10^−6^
No. clients last week	9 (0–50)	12 (0–70)	0.50
Self-reported HIV+ clients last week	1 (0–3)	1 (0–3)	0.52
Unprotected sex acts in the last 7 days	2 (0–5)	2 (0–12)	0.55
BV ^b^ status negative (*n*, %)	23, 100%	26, 100%	n.s.
Yeast infection negative (*n*, %)	0, 0%	0, 0%	n.s.
Vaginal douching (*n*, %) ^c^	23, 100%	26, 100%	n.s.

^a^ n.s.: not significant (*p* < 0.05), unpaired *t* test with Welch correction, two-stage step-up (Benjamin, Krieger, and Yekutieli); ^b^ BV: bacterial vaginosis with Nugent score ≥ 7. ^c^ Any douching performed by inserting water, or water and soap, in the vagina in the last 7 days.

**Table 2 viruses-14-00471-t002:** List of ISGs expressed in the unstimulated ex vivo PBMC.

**35 ISGs Were Detected in Unstimulated Ex Vivo PBMC ^a^**
*ACTB/B-ACTIN, ATM, BAX, BCL2, BclXL/BCL2L1, CCL5/RANTES, CDK4, CDKN2A/p16/p14, CXCL10/IP-10, DPP4/CD26, E2F1, FCGR2B, EIF2AK2/PKR, HDM2, IFI6/G1P3, IFIH1/MDA5, IFIT1/ISG56, IFIT3/ISG60, IFNA2, IFNA4, IFNB1, IL1B, IRF-1, IRF-7, IRF-9, ISG15, MXA, MYD88, PMAIP1/NOXA, PTGS2/COX2, RIGI/DDX58, STAT1, TNF/TNFA, TNFSF10/TRAIL, TNFRSF10A/TRAILR1*
**41 ISGs Were NOT Detected in Unstimulated Ex Vivo PBMC ^a^**
*5LO**, ANKRD1, BRCA1, C5, CAR, **CASP8**, CASP9, CDC25B, CDC25A, CDK1, CDK6, CDK2, CDK4A, CDK7, CIAP2, FAS, FASL, **FLIP**, FLAP, **GADD45**, GPT2, IFNa113, **IRF4***, ***IRF5****, LMP7, LMP10, MCL1, **MX2**, OAS1, P21cip1, **p53/Tp53**, p8/NUPR1, PELI1, PSME2, PUMA, RB1, RELA, RSAD2/Viperin **TAP1**, XIAP, ZFP36L2*

^a^ Cellular RNA transcript levels were quantitated using RT-PCR, and total transcript levels were normalized to 18S rRNA. Expression of ISGs were positive when the threshold cycle (Ct) is 2Ct > the negative controls (*n* = 16). Underlined ISGs were not detected in PBMCs stimulated with IFN-γ/IFN-α2; the induced ISGs (bolded) were trending but not statistically different between the two subgroups.

**Table 3 viruses-14-00471-t003:** Expression of ISGs in unstimulated HESN versus non-HESN PBMC ^a^.

HESN Only (0)	HESN & Non-HESN (25)	Non-HESN Only (10)
	*ACTB/B-ACTIN, BAX, **CXCL10/IP-**10, CCL5/RANTES, CDKN2A/p16/p14, DPP4/CD26, EIF2AK2/PKR, E2F1, HDM2, **IFI6/G1P3**, **IFIH1/MDA5**, **IFIT1/ISG56**, **IFIT3/ISG60**, IFNA4, **IFNB1**, **IRF-1**, IRF-7, IRF-9, **MX1**, MYD88, **PTGS2/COX2**, **TNF/TNFA**, **TNFSF10/TRAIL1**, **TNFRSF10A/TRAILR1**, STAT1*	*ATM, BCL2, BclXL/BCL2L1, CDK4, FCGR2B, IFNA2, IL-1B, ISG15, PMAIP1/NOXA, RIG-1/DDX58*

^a^ Cellular RNA transcript levels were quantitated using RT-PCR, and total transcript levels were normalized to 18S rRNA. Expression of ISGs were positive when the threshold cycle (Ct) is 2Ct > the negative controls (*n* = 7 in HESN group, *n* = 9 in the non-HESN control group).

## Data Availability

All data supporting our findings are found in this manuscript.

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
