# Peer review of "Transient Increases in Inflammation and Proapoptotic Potential Are Associated with the HESN Phenotype Observed in a Subgroup of Kenyan Female Sex Workers"

_viruses, 2022, doi:10.3390/v14030471_

Round 1
Reviewer 1 Report
This manuscript would benefit from the EQUATOR guidelines (e.g., STROBE). Currently, it is difficult to read and the main findings are not sufficiently discussed.
Overall, the study used rather unique samples, describes the expression of several IFN family genes and the expression of IFN-induced genes in two distinct populations of sex workers. It ends with an siRNA experiment targeting IRF-1 expression and compares ISG expression between the two populations. The data are of interest to the community.
Introduction: the overview of factors making person resistant to HIV infection is minimal.
M&M: The criteria to define a person resistant to HIV infection can be discussed, as well as whether the second group could be used as a “control group”. The criteria and the study design have their limitations that are not discussed in the manuscript including in the last part of the discussion.
Results:
Lane 150: “mucosal samples” is most probably an error. Never discuss before and after
Lane 150: “mucosal samples” is most probably an error. Never discuss before and after
Lane 170 and …: the rationale to restrict analysis to 35 ISG sounds very light. It should be also discussed in the limitation section.
Table 1: did the unprotected intercourses involved HIV positive clients? BV not define? Any information on other STI, HSV-2 infection? P value with 6 digits has no sense.
Renumber the table because there are two table 1.
Section 3.2 and following: all sections include a brief rational, brief M&M. There is no need to repeat this information if the introduction and M&M sections are properly written. Second, it is not necessary to repeat what can be found in the table, only the main observations. Finally, it is not common to read a sentence in this section beginning with "these results suggest that", this belongs more in the discussion.
Lane 180: “Contrary to the hypothesis…” this hypothesis was not developed before
Use the same abbreviations throughout the text, HESN or FSW HESN or HESN CSW.
Typos: lane 200, 213, 287 288, 292, ...
Author Response
Reviewer no. 1
This manuscript would benefit from the EQUATOR guidelines (e.g., STROBE). Currently, it is difficult to read and the main findings are not sufficiently discussed.
Response: The findings have been better organized into 3 major focuses and more discussion of the findings have now been included.
Overall, the study used rather unique samples, describes the expression of several IFN family genes and the expression of IFN-induced genes in two distinct populations of sex workers. It ends with an siRNA experiment targeting IRF-1 expression and compares ISG expression between the two populations. The data are of interest to the community.
Response: Thank you for recognizing the merits of this study.
Introduction: the overview of factors making person resistant to HIV infection is minimal.
Response: An overview of the factors that contributing to reduced HIV susceptibility is now included in the introduction.
M&M: The criteria to define a person resistant to HIV infection can be discussed, as well as whether the second group could be used as a “control group”.
Response: The criteria used to define HESN is further discussed in the M&M and in the ‘limitation’ section in the Discussion. The definition of the ‘non-HESN controls’ was also discussed in the M&M and the discussion. More epidemiological data on the ‘non-HESN controls’ is now included.
The criteria and the study design have their limitations that are not discussed in the manuscript including in the last part of the discussion.
Response: A limitation section is now included at the end of the discussion.
Results:
Lane 150: “mucosal samples” is most probably an error. Never discuss before and after
Response: It is now removed.
Lane 170 and …: the rationale to restrict analysis to 35 ISG sounds very light. It should be also discussed in the limitation section.
Response: The reasoning for focusing on the 35 ISGs was addressed in the result section and the potential caveats associated with focusing on only 35 ISGs were discussed in the limitation section in the Discussion.
Table 1: did the unprotected intercourses involved HIV positive clients? BV not define? Any information on other STI, HSV-2 infection? P value with 6 digits has no sense.
Response: Both the HESN and the non-HESN FSWs have reported having sex with HIV-positive partner (n > 1). Less than 70% of the HESN FSWs reported using condom and 100% of the non-HESN FSWs reported using condom during intercourse in the questionnaire. However, despite the reported use of condoms during sexual intercourse, seroconversion was observed in the non-HESN controls. BV is now defined. Detection of yeast infection and BV were part of the resurvey. Other STI were not tested. All participants in this study did not report any discomfort at the vaginal site. However, that does not exclude the possibility of vaginal HPV or HSV. Vaginal samples are available, should the study focus on the cervical vaginal mucosa.
Renumber the table because there are two table 1.
Response: Apologies. All tables and figures are now properly labelled.
Section 3.2 and following: all sections include a brief rational, brief M&M. There is no need to repeat this information if the introduction and M&M sections are properly written. Second, it is not necessary to repeat what can be found in the table, only the main observations.
Response: Repetition has been removed from the result session, which has been rewritten to better organize the data for presentation.
Finally, it is not common to read a sentence in this section beginning with "these results suggest that", this belongs more in the discussion.
Response: Thank you for the reminder. Elaboration on the findings has now been removed from the result section.
Lane 180: “Contrary to the hypothesis…” this hypothesis was not developed before
Response: It is now removed.
Use the same abbreviations throughout the text, HESN or FSW HESN or HESN CSW.
Typos: lane 200, 213, 287 288, 292, ...
Response: It is now corrected.
Reviewer 2 Report
In this manuscript by Gluchowsky et al. “Altered transcription profile of interferon-stimulated genes may play a role in maintaining the low susceptibility to HIV acquisition observed in a subgroup of Kenyan commercial sex workers” authors investigated the correlation between ISGs function and susceptibility to HIV infection in a cohort of female sex workers. The study found a different pattern of ISGs and activation of transcripts expression between HIV-exposed seropositive and seronegative women. Although the study shows interesting results, authors fail to provide a comprehensive view of the potential unique pattern associated to either profile. The title does not reflect the findings since the authors did not analyze directly whether such pattern is functionally associated to HIV acquisition susceptibility.
Specific comments and some typos that should be addressed are listed below.
Title: does not reflect the findings of the study.
Line 55-57: check the sentence
Lines 67-74: It is not clear what results have been published before and which ones are addressed in this manuscript. Please, review.
Line 85: define “FSW”
Table 1: define bacterial vaginosis “BV”. The table should follow the same structure and for BV and yeast infection, a number and frequency in parenthesis shown. i.e., 23 (100%). Clarify (line 154) whether they were all negative or all positive for BV and yeast infection in the table. Table 1 should be placed after line 161.
Line 153: typo: enrollment
Line 161: Add the difference on active in sex work time and clarify how can this affect the interpretation of the results.
Table 1 and 2 are mislabeled and confusing. They should read 2 and 3. Titles are confusing. I suggest trying to merge them into a single table. It is very difficult to read and interpret lines 162-183.
Lines 196-197: where are these results?
Place Figure 1 below line 197.
Line 198: Title, what are the authors referring to with “sub-optimally up-regulated? Sounds confusing
Lines 199-200: Did authors analyze whether the IRF-1 SNP was present in HESN? That sentence is assuming that whatever differences are detected next, are specifically due to that SNP. Please, modify accordingly.
Lines 198-240, Figure 2: Where are the 12 ISGs that had significant changes upon treatment? Authors should reorganize the figure to show up/downregulated transcripts showing IGs first, then Chemo and chemo receptors..etc… If authors are starting to describe the general effect, they should also comment on the apoptotic ones. It is very difficult to follow the text along with the figure. Lines 216-219 should be presented first (in line 207)…and so on. Reorganize the text to follow the same order shown in the figure. Alternatively, breaking the figure into 3 panels might be helpful where one shows up/down regulation of ISGs and other trnascripts, the second one comparison between the groups.
Line 213; typo, expression
Lines 244-246: Why did the authors chose those specific transcripts?
Line 251: typo CSW
Line 254: typos These result and, robust
Figure 3 legend is misplaced. Move the figure below line 260
Lines 254-256: The comparison about the robustness of the response is not accurate. Authors should reword highlighting the differences: HESN had a transient increase in the transcripts while non-HESN…Was expression of transcripts statistically significant between HESN and non-HESN at any time-point? Statistics are missing. The same apply to Figure 4. Authors should explain the importance of using PBMC and isolated CD4+ T cells.
Line 284: typo: is instead of was
Line 288: regulation typo
Figure 5: Figure B should be merged into one single panel and show current comparisons plus comparison of mock and IFN between HESN and non-HESN. Legend: The (A) should be moved to …”were evaluated (A) before and (B)…
Lines 295-317: The term robust is again confusing. The title could be simplified saying higher but transient. Statistics are missing.
Line 310: continuous typo
Line 326: Sub-title is too long and it should be clarified. The figure does not support this statement.
Line 334: why were those ISGs chosen and authors did not follow their own results? What is Figure 7 comparing? Where is the comparison between HESN and non-HESN that support the subheading of this paragraph? All these results are really confusing.
Author Response
Reviewer no. 2.
This study investigated the molecular mechanisms behind the low acquisition of HIV infection in a cohort of Kenyan female sex workers who remain seronegative despite exposure to HIV-1 (HESN). The authors hypothesized strong antiviral responses in HESN PBMCs and CD4+ T cells. This hypothesis was tested by measuring levels of ISGs with and without IFN treatment. Surprisingly, HESN PBMCs exhibited robust, but transient response to IFN stimulation, compared with non-HESN PBMCs. In addition, HESN PBMCs expressed higher levels of negative regulators of IFN response, which explains the transient nature of ISG expression. Interestingly, non-HESN cells express more pro-inflammatory and pro-apoptotic ISGs. The authors concluded that quickly resolving IFN response may protect against HIV acquisition.
It is interesting to note that, instead of performing RNAseq or single cell RNAseq, the authors chose to measure the expression levels of 76 selected ISGs by RT-PCR. Please discuss the potential limitation of measuring a selective group of genes compared with a more comprehensive analysis of global gene expression after IFN stimulation.
Response: At the time of the study, performing RNA-seq is quite costly. It was a decision of analyzing a few samples from each group and performing an analysis with more depth (i.e., kinetics, stimulation) or to perform a global analysis with a few samples. On a separate study, RNA-seq was performed with 4 PBMC samples from each of the HESN and the non-HESN group; only few ISG were found to be differentially expressed between the two group and most of those have been included in this study. The expression of ISGs is tightly regulated and specific to cell types; hence, level of ISG transcripts may be diluted in the RNA pool from PBMC. In addition, findings from the global analysis still require validation using RT-qPCR with validated primer set. There are more than a thousand of ISGs; hence, it is possible that several potential ISGs that are critical for the HESN phenotype were missed in this study and that data from this study is only a tip of an iceberg. However, findings in this study highlighted the importance of strict regulation of ISGs in the HESN phenotype and prompted further investigations into the mechanisms in regulation the transient ISG responses to IFN stimulation. This is now discussed in the limitations.
Out of the 76 ISGs examined, 35 were detected in the unstimulated PBMCs, and analysis was performed on these 35 genes. When measuring the effect of IFN stimulation, changes in these 35 genes, not the 76 ISGs, were determined. It is possible that, even though the other 41 ISGs were not detected in the unstimulated PBMCs, their expression might be turned on by IFN treatment. Please discuss this possibility and the potential impact on the conclusions made.
Response: Additional reasoning for focusing on the 35 ISGs has been included in Section 3.1. We agree with the reviewer. The other 41 ISGs, although not detected via RT-PCR in not-stimulated PBMC, might still be differentially regulated in the HESN PBMCs upon viral infection or challenge with exogenous stimuli. A pilot study was performed to examine the transcriptional responses of these 41 ISGs to a 3-hour of stimulation with exogenous interferon-g (IFN-g, 10ng/ml) and IFN-a2 (10ng/ml) in small subsets of PBMC samples (n=4, HESN and n=5 non-HESN). Similar increases in the transcripts of 36 ISGs (out of 41 ISGs examined) were found in both HESN and non-HESN groups (data not shown, Table 2: bolded ISGs). Among the 36 responsive ISGs, 8 ISGs (CASP8, FLIP, GADD45, IRF4, IRF5, MxB, p53, TAP1) showed trending differences between the 2 groups (12-38% difference in transcript levels, p-values ranged 0.065 to 0.09). Five ISGs remained undetected following stimulation with IFN-g and IFN-a2 (Table 2: underlined ISGs). As the differences in transcriptional responses to IFN-stimulation were not dramatically different between the HESN and the non-HESN PBMCs (12-38% in transcript level; (i.e., less than 2 fold changes), we do not expect significant impact on the conclusions made.
In this HESN cohort, HIV-1 acquisition is primarily through mucosal transmission, hence the main protection mechanism should act at the mucosal tissues. Please discuss how the findings with PBMCs can be applied to cut down HIV-1 infection at mucosa.
Response: Although there are significant discrepancies between the cells in the blood and the cells found at the mucosal compartments (Horton et al., PMID: 20011545), findings from studies of PBMCs are still valuable in guiding research at the vaginal site. In demonstrating the importance of strict regulation of ISGs in PBMCs, regulation of ISGs should be studied with mucosal cells. In fact, ongoing study in our group used CD4-AsiC that are specific for IRF-1 to induce IRF-1 knockdown in human CD4+ cervical mononuclear cells showed reduced susceptibility to HIV-1 infection by >90%. IRF-1 knockdown is also being tested in hBLT mice via intravaginal administration of CD4-AsiC. Mucosa is not a primary immune tissue, mucosal cells, especially innate immune cells are replenished via blood and ‘conditioned/educated’ by the mucosal environment. It should be further explored whether the intrinsic mechanisms of gene regulation are similar in PBMC and mucosal cells.
Line 217, PBMS should be PMBC. Fig 6 legend: the treatment should be IFN-alpha.
Response: It is now corrected.
Reviewer 3 Report
This study investigated the molecular mechanisms behind the low acquisition of HIV infection in a cohort of Kenyan female sex workers who remain seronegative despite exposure to HIV-1 (HESN). The authors hypothesized strong antiviral responses in HESN PBMCs and CD4+ T cells. This hypothesis was tested by measuring levels of ISGs with and without IFN treatment. Surprisingly, HESN PBMCs exhibited robust, but transient response to IFN stimulation, compared with non-HESN PBMCs. In addition, HESN PBMCs expressed higher levels of negative regulators of IFN response, which explains the transient nature of ISG expression. Interestingly, non-HESN cells express more pro-inflammatory and pro-apoptotic ISGs. The authors concluded that quickly resolving IFN response may protect against HIV acquisition.
It is interesting to note that, instead of performing RNAseq or single cell RNAseq, the authors chose to measure the expression levels of 76 selected ISGs by RT-PCR. Please discuss the potential limitation of measuring a selective group of genes compared with a more comprehensive analysis of global gene expression after IFN stimulation.
Out of the 76 ISGs examined, 35 were detected in the unstimulated PBMCs, and analysis was performed on these 35 genes. When measuring the effect of IFN stimulation, changes in these 35 genes, not the 76 ISGs, were determined. It is possible that, even though the other 41 ISGs were not detected in the unstimulated PBMCs, their expression might be turned on by IFN treatment. Please discuss this possibility and the potential impact on the conclusions made.
In this HESN cohort, HIV-1 acquisition is primarily through mucosal transmission, hence the main protection mechanism should act at the mucosal tissues. Please discuss how the findings with PBMCs can be applied to cut down HIV-1 infection at mucosa.
Line 217, PBMS should be PMBC.
Fig 6 legend: the treatment should be IFN-alpha.
Author Response
Reviewer no. 3
In this manuscript by Gluchowsky et al. “Altered transcription profile of interferon-stimulated genes may play a role in maintaining the low susceptibility to HIV acquisition observed in a subgroup of Kenyan commercial sex workers” authors investigated the correlation between ISGs function and susceptibility to HIV infection in a cohort of female sex workers. The study found a different pattern of ISGs and activation of transcripts expression between HIV-exposed seropositive and seronegative women.
Although the study shows interesting results, authors fail to provide a comprehensive view of the potential unique pattern associated to either profile. The title does not reflect the findings since the authors did not analyze directly whether such pattern is functionally associated to HIV acquisition susceptibility.
Response: The unique ISG profile is now clearly stated in the first paragraph of the Discussion. The title has been revised to remove ‘HIV acquisition susceptibility’, even though HESN FSWs do have reduced HIV acquisition susceptibility.
Specific comments and some typos that should be addressed are listed below.
Title: does not reflect the findings of the study.
Response: The title has been revised to remove ‘HIV acquisition susceptibility’
Line 55-57: check the sentence
Lines 67-74: It is not clear what results have been published before and which ones are addressed in this manuscript. Please, review.
Line 85: define “FSW”
Response: Those have been re-written or corrected. ‘FSW’
Table 1: define bacterial vaginosis “BV”. The table should follow the same structure and for BV and yeast infection, a number and frequency in parenthesis shown. i.e., 23 (100%). Clarify (line 154) whether they were all negative or all positive for BV and yeast infection in the table. Table 1 should be placed after line 161.
Response: ‘BV’ is now defined in the footnote of Table 1. The table is now revised according to the reviewer’s suggestion.
Line 153: typo: enrollment
Response: corrected.
Line 161: Add the difference on active in sex work time and clarify how can this affect the interpretation of the results.
Response: This is now discussed. On the note of environmental influences, the HESN FSWs were in the sex trade for a significantly longer periods of time and had been exposed to more clients; it is highly possible that the HESN phenotype was driven by high exposure to foreign antigens, which induced immune tolerance to avoid inflammation at the mucosal site. However, the mechanism(s) of such remained to be sought. As IRF-1 regulates the differentiation of regulatory T cells, via suppressing FOXP3 trans-activation, immune tolerance could be induced by reducing IRF-1 expression. This should not affect the interpretation of the results.
Table 1 and 2 are mislabeled and confusing. They should read 2 and 3. Titles are confusing. I suggest trying to merge them into a single table. It is very difficult to read and interpret lines 162-183.
Response: Both tables are revised and described better in the text. The subtitles have also revised as the findings were re-organized for a clearer presentation.
Lines 196-197: where are these results?
Place Figure 1 below line 197.
Line 198: Title, what are the authors referring to with “sub-optimally up-regulated? Sounds confusing
Response: There has been great technical difficulties in keeping the figures and tables in place when trying to fit the manuscript in the Viruses format. Apologies for the confusion. It is now fixed. ‘sub-optimally’ was removed form the text.
Lines 199-200: Did authors analyze whether the IRF-1 SNP was present in HESN? That sentence is assuming that whatever differences are detected next, are specifically due to that SNP. Please, modify accordingly.
Response: Thank you for the reminder. Some of these HESN FSWs (n=18) have IRF-1 SNP. This section is now corrected.
Lines 198-240, Figure 2: Where are the 12 ISGs that had significant changes upon treatment?
Response: Those are the 12 out of the 16 ISGs shown in Figure 2. Those were now better described in the text.
Authors should reorganize the figure to show up/downregulated transcripts showing IGs first, then Chemo and chemo receptors..etc…
Response: We have reorganized a few figures and the analytes and added labels for a clearer presentation. Hopefully, we have responded adequately to the reviewer's suggestions.
If authors are starting to describe the general effect, they should also comment on the apoptotic ones.
Response: Pro-apoptotic ISGs expression and response to IFN stimulation have been expanded in the results and discussion.
It is very difficult to follow the text along with the figure. Lines 216-219 should be presented first (in line 207)…and so on.
Reorganize the text to follow the same order shown in the figure. Alternatively, breaking the figure into 3 panels might be helpful where one shows up/down regulation of ISGs and other trnascripts, the second one comparison between the groups.
Response: We apologize for the poor writing. Major revision in the writing has been done to make the presentation easier to follow. And thank you for your detailed comments to help guide the revision of this manuscript to a much better version.
Line 213; typo, expression
Response: it is now corrected.
Lines 244-246: Why did the authors chose those specific transcripts?
Response: Reasoning for choosing these specific transcripts is expanded in section 3.1 and in the limitation section, at the end of the Discussion. These 76 ISGs have been studied in numerous antiviral response. Huge amounts of efforts were made in designing and validating the primer pairs that specifically amplify these 76 ISGs with high specificity and similar efficiencies allowing comparisons of transcript levels between different ISGs.
Line 251: typo CSW
Line 254: typos These result and, robust
Figure 3 legend is misplaced. Move the figure below line 260
Response: it is now corrected.
Lines 254-256: The comparison about the robustness of the response is not accurate. Authors should reword highlighting the differences: HESN had a transient increase in the transcripts while non-HESN…Was expression of transcripts statistically significant between HESN and non-HESN at any time-point? Statistics are missing. The same apply to Figure 4.
Response: The writing has been corrected. And yes, t-tests were used at each time point to determine whether there is a difference between the 2 groups.
Authors should explain the importance of using PBMC and isolated CD4+ T cells.
Response: “While assessing ISG expression and response in PBMC enabled the evaluation of systematic ISG regulation, representing a general antiviral response, assessing ISG regulation in the CD4+ T cells, allowed the evaluation of immune activation of the main HIV targets. This is of importance because HIV-1 replicates in activated CD4+ T cells.” This is now added to section 3.4.
Line 284: typo: is instead of was
Line 288: regulation typo
Response: Correction has been made.
Figure 5: Figure B should be merged into one single panel and show current comparisons plus comparison of mock and IFN between HESN and non-HESN. Legend: The (A) should be moved to …”were evaluated (A) before and (B)…
Response: Thanks for the suggestion to merge the two figures in panel (B). We have tried the suggestion but decided to keep the two graphs separate. As merging the two graphs in (B) repeated the information in (A) and make the description of data more complicated. Fold changes in SOCS-1 response to IFN-γ is presented in (C).
Lines 295-317: The term robust is again confusing. The title could be simplified saying higher but transient. Statistics are missing.
Line 310: continuous typo
Line 326: Sub-title is too long and it should be clarified. The figure does not support this statement.
Response: It is now corrected. We have removed the word ‘robust’. It describes a quick increase in transcribe levels in the HESN, when compared to the slow/steady increase of transcripts in the non-HESN control. The sub-title is also revised.
Line 334: why were those ISGs chosen and authors did not follow their own results?
Response: Reasoning for choosing these specific transcripts is expanded in section 3.1 and in the limitation section, at the end of the Discussion. These 76 ISGs have been studied in numerous antiviral response. Huge amounts of efforts were made in designing and validating the primer pairs that specifically amplify these 76 ISGs with high specificity and similar efficiencies allowing comparisons of transcript levels between different ISGs.
What is Figure 7 comparing?
Response: Figure 7 examined the effects of knocking down IRF-1 expression on the baseline expression of ISGs in CD4+ T cells. The expression of ISGs in cells (from non-HESN control) treated with IRF-1 specific CD4-AsiC was compared to the ISG expression in cells treated with control CD4-AsiC.
Where is the comparison between HESN and non-HESN that support the subheading of this paragraph? All these results are really confusing.
Response: Apologies for the confusion. Figure 7 used only CD4+ T cells from the non-HESN FSWs. The results showed that these CD4+ T cells with IRF-1 knockdown (reduced IRF-1 expression) had reduced baseline level of MX1 and IRF-7 (pro-inflammatory ISGs) but increased baseline level of TRAILR1 (pro-apoptotic ISGs). This ISG profile is similar to that found in the HESN PBMC and CD4 T cells. This is compared to the CD4+ T cells without IRF1 knockdown which have the ISG expression profile similar to that of CD4 T cells from the non-HESN control. The writing has been revised to explain this better.
Round 2
Reviewer 2 Report
This revised version of the manuscript has significantly improved. Results are presented in a clear way and conclusions are supported by the results.